



# A Bayesian model for quantifying errors in citizen science data: application to rainfall observations from Nepal

Jessica A. Eisma[1], Gerrit Schoups[2], Jeffrey C. Davids[3,4], and Nick van de Giesen[2]

[1]University of Texas at Arlington, 701 S Nedderman Dr., Arlington, TX, 76019, USA
[2]Delft University of Technology, Mekelweg 5, Delft, 2628 CD, The Netherlands
[3]California State University, Chico, 400 W 1st St, Chico, CA, 95929, USA
[4]Davids Engineering, 1095 Nelson Street, Chico, CA, 95926, USA

**Correspondence:** Jessica A. Eisma (jessica.eisma@uta.edu)

**Abstract.** High quality citizen science data can be instrumental in advancing science toward new discoveries and a deeper understanding of under-observed phenomena. However, the error structure of citizen scientist (CS) data must be well-defined. Within a citizen science program, the errors in submitted observations vary, and their occurrence may depend on CS-specific characteristics. This study develops a graphical Bayesian inference model of error types in CS data. The model assumes that:

(1) each CS observation is subject to a specific error type, each with its own bias and noise; and (2) an observation's error type depends on the error community of the CS, which in turn relates to characteristics of the CS submitting the observation. Given a set of CS observations and corresponding ground-truth values, the model can be calibrated for a specific application, yielding (i) number of error types and error communities, (ii) bias and noise for each error type, (iii) error distribution of each error community, and (iv) the error community to which each CS belongs. The model, applied to Nepal CS rainfall observations,

identifies five error types and sorts CSs into four model-inferred communities. In the case study, 73% of CSs submitted data with errors in fewer than 5% of their observations. The remaining CSs submitted data with unit, meniscus, unknown, and outlier errors. A CS's assigned community, coupled with model-inferred error probabilities, can identify observations that require verification. With such a system, the onus of validating CS data is partially transferred from human effort to machine-learned algorithms.

## 1 Introduction

Communities worldwide face increasing uncertainty regarding extreme weather events due to climate change. Reliable weather forecasts allow a community to initiate proactive measures when anticipating an extreme event—measures that sometimes save hundreds, if not thousands of lives. Unfortunately, sparse weather data in many regions of the world inhibit coordinated response efforts of local and regional governments (Teague and Gallicchio, 2017, p. 218). Further, some countries lack the

institutional capacity to effectively utilize available weather data, inhibiting the generation and distribution of locally relevant weather forecasts. Citizen science can help bridge such data gaps.

Citizen science programs, organized efforts to collect scientific data in collaboration with members of the public, have become increasingly popular as advances in technology have made the data collection and submission process more accessible



(Bonney et al., 2009; Newman et al., 2012). However, some traditional scientists, policymakers, and workers in federal water
bureaucracies continue to question the quality of data submitted by members of the public, and have yet to accept the legiti-
macy of scientific discoveries advanced by citizen scientists (CSs) (Hunter et al., 2013; Paul et al., 2018; Riesch and Potter,
2014; Sheppard and Terveen, 2011). Others, however, have embraced citizen science as an effective means for increasing the
spatiotemporal resolution of scientific data while acknowledging the data may contain errors. Many citizen science programs
investigate the type and frequency of mistakes in the data collected by program participants and develop training initiatives
designed to reduce errors (Bird et al., 2014; Crall et al., 2011; Davids et al., 2019). While mistakes in citizen science data are
well-recognized, issues with incompleteness, data gaps, and fragmentary recording may also limit the utility of citizen science
data (Paul et al., 2020).

Most CS programs conduct quality control of the data submitted by their participants, but the time and effort invested varies
widely. For example, CSs report when they feel an earthquake and rank its strength for the United States Geological Survey's
(USGS) Did You Feel It? program. The USGS removes outliers and aggregates reported intensities at zip code or city-level
after processing the data through the Community Decimal Intensity algorithm (Atkinson and Wald, 2007). On the other end
of the spectrum, CS programs like SmartPhones4Water-Nepal (S4W-Nepal) undertake an intensive quality control process.
CSs submit rainfall depth observations to S4W-Nepal, and S4W-Nepal checks the value of each submitted rainfall observation
against an accompanying photograph of the rain gauge and manually corrects erroneous observations (Davids et al., 2019).
The range of time and effort dedicated to conduct quality control for citizen science data varies greatly across programs.

Most error analyses of citizen science data focus on identifying and removing outliers from a dataset. Trained filters flag
outliers by identifying observations that do not fit within the expected range of values or classes, such as species range or
allowable count (Bonter and Cooper, 2012; Wiggins et al., 2011). Some citizen science programs develop eligibility or trust
rating procedures to identify users that are likely to submit correct observations (Delaney et al., 2008; Hunter et al., 2013).
Ratings schemes that consider demographic and experience-related characteristics have potential for describing the variability
in citizen science data reliability (Kosmala et al., 2016). However, some individual CSs do not submit enough observations to
be accurately assigned a rating. To overcome such limitations, Venanzi et al. (2014) employed model-based machine learning
to group CSs into four communities, each with a distinct pattern of errors.

Machine learning algorithms in the form of hierarchical, generalized linear, and mixed-effects models have been employed
by a variety of citizen science programs to study errors in citizen science data (Bird et al., 2014; Venanzi et al., 2014). Gen-
eralized linear models (GLMs) have largely been used to study whether and how characteristics of CSs affect the accuracy of
their observations (Butt et al., 2013; Crall et al., 2011; Delaney et al., 2008). GLMs can determine whether CS characteristics
significantly impact the likelihood of making a mistake, but they cannot infer the types of errors made. Mixed-effects models
add a random-effects factor to generalized linear models, permitting the study of errors in relation to an unintended grouping
effect, such as spatial clustering (Bird et al., 2014; Brunsdon and Comber, 2012). Mixed-effects models may effectively group
CSs into communities with similar characteristics or mistake tendencies, but, as with GLMs, they cannot quantify the number
and type of mistakes made. Alternatively, hierarchical models have been leveraged to study how CS mistakes relate to effort
and site-level effects (de Solla et al., 2005; Fink et al., 2010; Miller et al., 2011). Most machine learning-based models have





been used to study errors in qualitative citizen science data, such as species identification and labeling tweets (Cox et al., 2012;

Lukyanenko et al., 2019; Venanzi et al., 2014). Thus far, no error-based investigations of CS observations have developed a unified methodology that can both infer the number and types of errors present in quantitative data and group the CSs into communities based on mistake tendency and characteristics. Error modeling has only been employed to identify erroneous citizen science observations for quantitative data in a limited manner. In addition, most machine learning citizen science research has focused on datasets that are relatively static or slow-moving in the fields of biology and conservation (Lukyanenko

et al., 2019). To our knowledge, the study presented here is the first attempt to leverage hierarchical machine learning to assess errors in quantitative citizen science data with high spatiotemporal variability. Despite the range of existing research on citizen science errors, widely adaptable methods for analyzing errors in quantitative citizen science data remain largely unexplored.

Motivated by the need to reduce the time-cost for quality control of citizen science data without sacrificing effectiveness, this study seeks to develop a reliable, semi-automated method for identifying citizen science observations that require additional

verification. The objective is to improve quality control of quantitative citizen science data by developing a Bayesian inference model that discovers, explains, and possibly corrects the errors in observations submitted by CSs. The following research questions will be explored:

1. How can the type and magnitude of citizen science data errors be automatically identified from citizen science data and corresponding ground truth?

2. Given a calibrated model, to what extent can errors be detected and corrected without ground truth?

3. To what extent do CS characteristics help in identifying and screening errors?

A probabilistic graphical model was developed to address these questions based on assumptions about the probabilistic relationships between CSs, their characteristics, and types and magnitude of their errors. Here, CS characteristics means any additional information a citizen science program has about participating CSs, e.g., age, education, and occupation. The prob-

abilistic graphical model design (research question 1) includes a mixture of linear regressions sub-models relating true and observed values and includes an unknown number of linear regressions. The model also includes a probabilistic sub-model relating CS characteristics to error types. Each CS is grouped into a community based on their characteristics and error profile. Each community is characterized by a distinct distribution of error types which indicates the likelihood that a submitted observation should be reviewed further. The model was applied to investigate its utility, including the capabilities of the model in

identifying erroneous observations and predicting the true value of submitted observations and the impact of multiple observations of a single event on model performance (research question 2) and model performance when observations are submitted by CSs with unknown characteristics (research question 3).



## 2 Model development and implementation

### 2.1 Model approach

The model described here is based on the Community Bayesian Classifier Combination (CommunityBCC) Model developed in Venanzi et al. (2014). Whereas the CommunityBCC Model was initially developed to assess errors in crowdscourced tweet labels by grouping participants into communities of similar labelling tendencies, the model presented here extends the CommunityBCC Model to assess mistakes in CS-submitted rainfall measurements by grouping CSs into communities of similar characteristics and mistake tendencies. The decision to include CS characteristics in the community-assignment process was

motivated by past studies that identified a significant relationship between CS characteristics and performance (Crall et al., 2011; Delaney et al., 2008; Sunde and Jessen, 2013). The use of communities to generalize the reliability of quantitative CS observations is novel, as is the effort to correct those observations based on the overarching mistake tendencies of the CSs inferred community.

### 2.2 Assumptions and model structure

A Bayesian probabilistic graphical model was developed based on a number of assumptions about the data being modeled. These assumptions were used to inform the relationships between the variables and ensure the model accurately represents the modeler's understanding of the physical processes that underlie the data (Krapu and Borsuk, 2019; Winn et al., 2020). The following assumptions informed the development of the citizen science errors inference model:

1. Each CS belongs to a single community.

2. CSs in the same community will have similar demographic and experience-related characteristics and will have made similar types and frequencies of errors in prior submissions.

3. Each CS in a particular community always submits an observation with a community-specific error type distribution.

4. Each CS observation relates to an underlying true value with a systematic bias and random noise level that depends on the error type of the observation.

While the tendency of CSs to make mistakes may change as they gain experience, the model developed here assumes that a CS will not change communities over time. This simplifies the model while also including the potential impact of experience as a citizen characteristic. CS demographic information was assumed to be a factor in determining community, because demographics, such as age, experience, and education, are a useful predictor in CS performance (Crall et al., 2011; Delaney et al., 2008; Sunde and Jessen, 2013). As an imprint of a CS's lived experience, demographics may influence CS

performance. For example, education and occupation imply specific skills training, which may present in the CS's observation tendencies. In addition, factors like motivation and recruitment may impact a CS's dedication to collecting and reporting accurate observations. Notably, motivation and recruitment method were predictive factors in CS participation rate (Davids et al., 2019). The predictive power of demographics in determining community will be assessed.



These assumptions are translated into the following set of equations describing the probabilistic relationship between model variables. The terminology and symbology used here is based on probabilistic graphical models (Winn et al., 2020). We first state the main statistical relations used in the model and provide clarifications for the wider research community. The model equations are presented following the order of hierarchy in the model, i.e., equations for the observed data layers are shown first, followed by the equations for variables in each previous layer.

*Notation.* Consider there are $S$ CSs with $C$ characteristics submitting an $O$ observation with $\varepsilon$ latent error type for event $e$. Let $\vartheta_e$ be the latent true value of rainfall for the submitted observation, $O_{s,e}$ for event $e$. We use a lowercase subscript to denote an index (e.g. $\vartheta_e$ indicates there is a TrueValue variable for each event $e$). Greek letters represent latent (inferred) variables, and Latin letters represent observable variables. To keep notation simple, we assume a dense set of labels in which all CSs observe all events. However, model implementation does not require CSs to submit observations for all events, as in Venanzi et al. (2014).

We quantify systematic (bias) and random (noise) differences between observations and underlying true values by means of a linear regression model parameterized by an error-type specific slope $\alpha$, offset $\beta$ and precision (inverse variance) $\tau$:

$$O_{s,e}|\vartheta_e \sim \prod_{n=1}^{N} \mathcal{N}(\alpha_n \vartheta_e + \beta_n, \tau_n)^{\delta(\varepsilon_{s,e}-n)}, \tag{1}$$

where $O_{s,e}$ represents the observed amount of rainfall in event $e$ submitted by CS $s$, and $\vartheta_e$ is the corresponding true rainfall amount for event $e$. Given the TrueValue, $\vartheta_e$, of an observation, the observed value is thus generated from the product of $N$ Gaussian distributions with mean equal to an error-type specific linear function of the true value and an error-type specific variance, where $N$ is the number of error types. $\alpha$, $\beta$, and $\tau$ depend on error type $\varepsilon_{s,e}$. The Dirac delta function $\delta()$ in the exponent is used to mathematically represent the mixture of linear regressions (i.e. the gate in Fig. 1), as documented in Minka and Winn (2008). It follows that unconditionally, i.e. without knowing the error type, the relation between observed and true value is a mixture of error-type specific Gaussian distributions, with the weight of each Gaussian distribution in the mixture given by the probability of the corresponding error type.

Equation 2, below, describes the conditional probability table for each error type and community. The error type $\varepsilon_{s,e}$ of event $e$ observed by CS $s$ is assumed to be generated from a discrete distribution denoted by $Dis$ that depends on the community-specific probability vector $\mathbf{PErr}_\gamma$ and the community $\gamma_s$ that the CS belongs to:

$$\varepsilon_{s,e}|\mathbf{PErr}[\gamma_s] \sim Dis(\varepsilon_{s,e}|\mathbf{PErr}[\gamma_s]), \tag{2}$$

Similarly, the community $\gamma$ to which CS $s$ belongs is a discrete random variable generated from a discrete distribution that depends on the probability vector $\mathbf{PCom}_s$, which specifies the prior probability of CS $s$ belonging to each community:

$$\gamma_s|\mathbf{PCom}_s \sim Dis(\gamma_s|\mathbf{PCom}_s), \tag{3}$$





The value $Z_{c,s}$ of citizen characteristic $c$ for CS $s$ is generated from a discrete distribution that depends on the probability $PChar$, which is derived from the characteristic $c$ under consideration and the community $\gamma_s$ the CS belongs to:

$$Z_{c,s}|PChar_c[\gamma_s] \sim Dis(Z_{c,s}|PChar_c[\gamma_s]), \tag{4}$$

Equation 4 quantifies the probabilistic relationship between each citizen characteristic and each assigned community in the form of a conditional probability table. As seen in Equations 2-4, the model assigns each CS to a single community, automatically grouping CSs with similar characteristics and error tendencies.

Finally, the model is completed by specifying priors for the regression parameters ($\alpha$, $\beta$, $\tau$), the probability vectors (**PCom**, **PChar$_c$**, **PErr**), and $\vartheta_e$, given in Appendix A. The priors were different for the training and testing phases and are detailed in Section 3.4. Generally, the training phase priors for **PCom**, **PChar$_c$** and **PErr** are uniform Dirichlet distributions, and the training phase priors for $\alpha$, $\beta$, $\tau$, and $\vartheta_e$ are Gaussian distributions with mean and variance informed by the testing data. The testing phase priors are equal to the training phase posteriors for all latent variables.

## 2.3 Representation as a factor graph

Equations 1-4 are translated into a factor graph as shown in Figure 1. The factor graph describes the joint posterior probability of the model (see Equation A5), while omitting the prior distributions for the sake of clarity. The factor graph includes observable and latent (inferred) variables, factor nodes, edges (arrows), plates, and gates. Variables are depicted by shaded or unfilled ellipses. A shaded variable is an observable value; an unfilled variable is a latent value. Factor nodes are the small black boxes connected to variables, describing the relation between variables connected to the factor. Edges (directional arrows) connect factor nodes to variables (Winn et al., 2020).

*Plates.* Plates are the large boxes outlined in gray surrounding portions of the factor graph. Plates are a simplified way to express repeated structures. The number of times a structure will be repeated is based on the index variable shown in the bottom right corner of the plate (Winn et al., 2020). For example, in Figure 1, the structure within the characteristics plate is repeated $X$ times, where $X$ is equal to the number of CS characteristics the model considers.

*Gates.* Gates are indicated by a dashed box, as seen around the Regression factor node in Figure 1. Gates essentially act as a switch, turning on and off depending on the value of the selector variable, which is the error type here (Minka and Winn, 2008). When gates are used to define a distribution, that distribution is a mixture.

## 2.4 Model implementation

We implemented the probabilistic model using Microsoft Research's open source Infer.NET software framework (Minka et al., 2018). The Infer.NET framework provides adaptable tools to develop and run Bayesian inference for probabilistic graphical models. The modeler must define the variables, the dependencies between variables, and provide prior distributions for the variables that will be inferred.





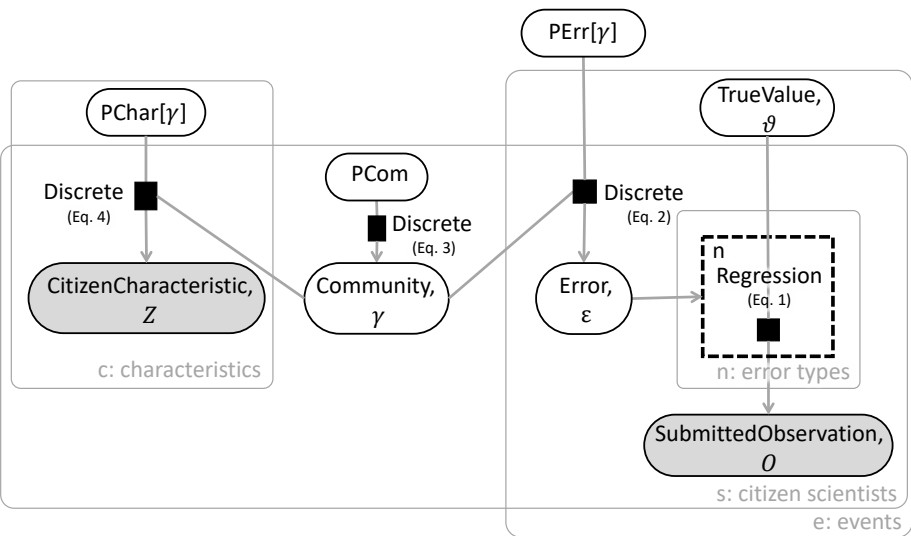

**Figure 1.** The citizen science error model depicted as a factor graph. A factor node represents a probabilistic relation between variables in the model and is shown by a black square. A variable is shown in an oval, with shading identifying observable variables. Arrows depict the output variable of each factor. A gate is represented by a dashed box. Plates are represented by gray rectangles with rounded corners. Symbols adopted from Winn et al. (2020).

Infer.NET generates a computationally efficient code for the inference algorithm using one of three available inference algorithms: expectation propagation, variational message passing, and Gibbs sampling. The model developed here employs the expectation propagation algorithm, because it is time efficient but reasonably accurate (Minka, 2013). Expectation propagation is a deterministic approximate inference algorithm for computing the marginal posterior distribution of each variable in the model (Minka, 2013). Each posterior distribution is assumed to take a specific parametric form in an exponential family (e.g. Gaussian, Gamma, discrete). The algorithm then aims to find parameter values for each parametric posterior that result in a good approximation of the exact posterior in terms of moment matching. For example, for a Gaussian approximation, expectation propagation will find a Gaussian whose mean and variance approximate those of the actual posterior. This is done using an iterative approach that starts from an initial guess for the approximate posteriors, and iteratively refines each posterior in turn via moment matching. Since all individual posterior updates depend on each other, the algorithm is iterated until all updates and posteriors stabilize (here, in <5 iterations). The final posteriors are not necessarily unique and may depend on how the algorithm was initialized. Here, we adopt a random initialization strategy for mixture models as used in Nishihara et al. (2013) and Minka et al. (2018) and evaluate non-uniqueness in the inferred posteriors using multiple runs with different random initialization.




## 3 Model application

Rainfall observations submitted by CSs have immense potential to increase the scientific community's understanding of rain events which are, by nature, highly heterogeneous in space and time. Currently, only about 1.6% of the land surface on Earth lies within 10 km of a rain gauge, and rain gauges are notoriously inconsistent (Kidd et al., 2017). So much so that the correlation coefficient for rain gauges 4 km apart in the midwestern United States was less than 0.5 for instantaneous rainfall (Habib et al., 2001). Citizen science rainfall observation programs must contend with the systematic errors inherent in measuring rainfall, as well as the errors induced by the CSs. Detailed investigations into the errors made by CSs, such as the efforts of S4W-Nepal, can help increase the utility of citizen science data and inform future program development, and is the subject of this study.

### 3.1 Study area

SmartPhones4Water Nepal (S4W-Nepal) partners with CSs across Nepal to collect rainfall observations (see Figure 2). Across Nepal, rainfall is highly heterogeneous in space and time. Average annual rainfall in Nepal varies from 250 mm on the leeward side of the Himalayas to over 3,000 mm in the center of the country near Pokhara (Figure 2) (Nayava, 1974). The South Asian summer monsoon brings approximately 80% of Nepal's annual precipitation during the months of June to September (Nayava, 1974). The majority of CSs participating in S4W-Nepal's rainfall data collection efforts reside in the Kathmandu Valley, home to about 10% of Nepal's population (Vibhāga, 2012). While the average annual precipitation is approximately 1,500 mm in the city of Kathmandu and 1,800 mm in the surrounding hills, it is highly variable and unpredictable (Thapa et al., 2017).

### 3.2 Data

S4W-Nepal recruits CSs to participate in a crowdsourced rainfall observation program in Nepal. S4W-Nepal collects the submitted observations via the Open Data Kit application for smart phones. Submitted observations include geo-location data, time of measurement, CS-reported depth of rainfall in millimeters, and a photograph of the rain gauge (Davids et al., 2019). The program is ongoing and has collected over 24,500 observations from over 265 CSs since 2016. An overview of the S4W-Nepal data is provided below; a detailed description can be found in Davids et al. (2019).

#### 3.2.1 Rain gauges

The participants were given a rain gauge constructed by S4W-Nepal and provided instructions on the proper installation and recording of rainfall data. The rain gauges were constructed from a re-purposed clear plastic bottle with a 100 mm diameter. The bottle was filled with a few centimeters of concrete to provide stability and a level measuring surface. The lid of the bottle was cut off where the taper ends, inverted, and placed flush with the top of the bottle to reduce evaporation losses. Finally, a ruler with millimeter precision was attached to the bottle to assist the reading of the rainfall depth (Davids et al., 2019).





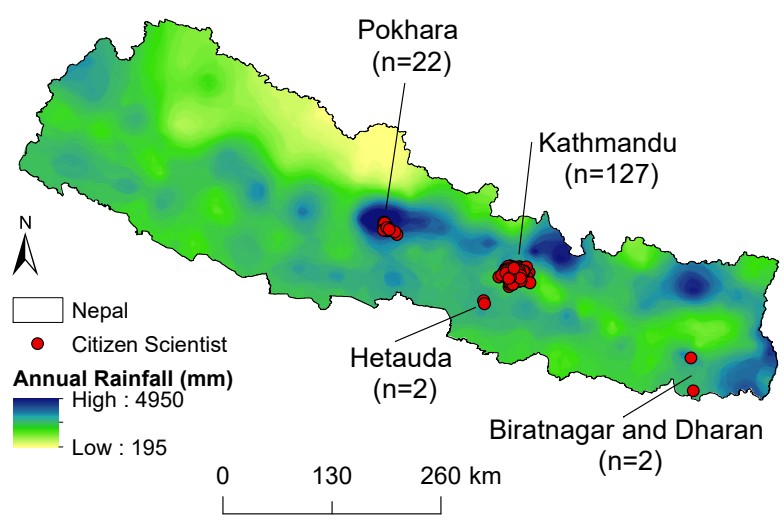

**Figure 2.** Locations of CSs for which characteristics are known with the number of CSs at specified locations shown in parentheses. Average annual rainfall grid created from observed data at 200 weather stations from 1980-2000 (USAID Nepal, 2013).

### 3.2.2 Citizen characteristics

During the recruitment process, S4W-Nepal recorded characteristic data for 153 CSs. Characteristics recorded were: motivation (paid/volunteer), recruitment method (personal connection, random site visit, social media, outreach), age ($\leq$18, 19-25, >25), education (<Bachelors, Bachelors, >Bachelors), place of residence (urban, semi-urban, rural), occupation (agriculture, student, other), and gender (male, female). CS characteristics will be used here to relate individual CSs with the likelihood of mistakes in the data they submit. All CS characteristics recorded by S4W-Nepal, regardless of pre-existing evidence that a characteristic is significantly correlated with CS performance, are included in the model. The model will determine the relative importance of each CS characteristic in defining mistake tendencies while inferring the community groups.

### 3.2.3 Erroneous observations

To detect erroneous rainfall observations submitted by CSs, S4W-Nepal checks the value of each submitted rainfall observation against the accompanying rain gauge photograph. If they detect an error, the correct rain depth is recorded while preserving the record of the original value submitted by the CS. This allows S4W-Nepal to track the types and frequencies of errors made by the CSs. Overall, approximately 9% of submitted rainfall observations are erroneous. Meniscus errors are the most common



(58% of errors; records capillary rise), followed by unknown errors (33%), and unit errors (8%; records data in centimeters rather than millimeters) (Davids et al., 2019).

### 3.3 Community and error selection

To select the appropriate number of communities to capture the differences among the CSs, model evidence was used. Model evidence indicates which model best explains the data relative to the model's complexity (MacKay, 2003, p. 343-386). While the model evidence is notoriously hard to compute, expectation propagation provides a convenient estimate as a by-product of its posterior approximations. Model evidence calculation in Infer.NET is achieved by inferring posterior component weights of a mixture consisting of two components, i.e. the entire model and the empty model (Minka, 2000).

Too many communities may lead to overfitting, whereas too few communities may lead to underfitting. The model evidence automatically makes this trade-off and identifies the optimal number of communities. Model evidence was computed for models with one to ten communities. The number of communities that resulted in the largest model evidence was selected as the correct number of communities for the model and data. Similarly, model evidence was used to determine how many error types were present in the data. Model evidence was computed for two to ten error types while using the optimal number of communities. The number of error types that resulted in the largest model evidence was selected as the number of error types for the model and data. After selecting the number of error types, model evidence was again checked to verify that the optimal number of communities remained constant. Selecting the error types via model evidence may identify more error types than expected, but the Bayesian model accounts for all possibilities and selects the one that most accurately represents the data.

### 3.4 Training and testing the model

Before training and testing, an additional assumption was incorporated, due to the nature of rainfall data: the inferred true value of rainfall was assumed to be between 0 and 540 mm. Rainfall events cannot result in negative rainfall, and 540 mm is the maximum one-day rainfall recorded for Nepal. Similar assumptions unique to a specific type of citizen science observation may be necessary at this stage of model development for application to other citizen science programs.

The inference model was trained and tested to ensure model performance was consistent across different groups of data. During training and testing, the following characteristics were known for each CS: motivation, recruitment, age, education, place of residence, occupation, gender, performance, and experience. The first seven characteristics were recorded by S4W-Nepal (as explained in Section 3.2). The last two characteristics, performance and experience, were defined based on the observations submitted by each CS. Performance is simply the percentage of observations submitted by a CS that did not require correction. A performance of 90% indicates that 90% of that CS's submitted observations matched the true value shown in the associated photograph. Experience is a count of how many observations a CS submitted through the 2018 monsoon season. Performance and experience rates were split into three levels based on natural breakpoints in their respective histograms.





### 3.4.1 Splitting the data

Rainfall observations submitted by CSs with known characteristics from 2016 to 2018 were randomly split into a training
data set and a testing data set. The training set consisted of 92% of available observations, representing 6,091 observations
submitted by 152 CSs. The CSs in the training set submitted anywhere from 1 to 159 observations, with the average number
of submissions being 43.5. The testing set consisted of the remaining 8% of available observations, representing 527 observa-
tions from 109 CSs. The CSs in the testing set submitted anywhere from 1 to 159 observations, with the average number of
submissions being 57.4. All CSs in the testing set were also in the training set. Note that individual observations in each group
were unique.

### 3.4.2 Training the model

Before training the model, prior distributions were set for the variables that were inferred. Uniform prior distributions were
set for the citizen characteristics, community, and error. The prior distribution for the true value parameter was a Gaussian
distribution with a mean equal to the average value of all submitted observations (15) and the four times the variance of the
entire dataset (2400; see Equation A1).

A true value prior variance of 2400 was chosen to reduce small event bias and accommodate inference of large rainfall
observations. The prior distributions for the Gaussian mixture parameters ($\alpha$, $\beta$, and $\tau$) were assigned based on the magnitude
of unit, meniscus, and unknown errors classified by Davids et al. (2019).

While running the model in the training phase, the characteristics for each CS, the submitted observations, and the true
values were known. The community for each CS, the error type for each submitted observation, the conditional probability
tables for each characteristic and error type, and parameters for the Gaussian mixture were inferred (see Equations 1-4 and
Figure 1). The training phase provided posterior distributions that were then used while testing the model.

### 3.4.3 Testing the model

To test the model, prior distributions for latent variables were set to the associated posterior distribution calculated during
training. The values of the submitted observations were set. The model inferred the community for each CS, the probable
error type for each observation, and provided a posterior distribution for the true value of the submitted observation. The
performance of the model was assessed based on whether the inferred posterior distribution for true value ($\vartheta$) covered the
true value identified in the accompanying photograph submitted by the CS and whether the mode of the true value posterior
matched the actual true value.

A synthetic rainfall event was created to explore how many observations of a single event are needed to produce a reliable
estimate of the event's true value. A synthetic observation of the event was created by first assigning an error type to each
CS based on the distribution of errors for their respective error communities (see Table 2). Then, the value of the synthetic
observation was calculated using Equation 1, the $\alpha$, $\beta$, and $\tau$ values from Table 1 with a true value of 15 mm. Multiple



synthetic events were created with two to three observations of the same event with one to two erroneous observations per
event. The true value of each synthetic event was predicted by the model.

## 4   Results and discussion

### 4.1   Sensitivity of $\alpha$, $\beta$, and $\tau$ priors and algorithm initialization

In the model application examined here, Davids et al. (2019) provided prior information on the types of errors in the data, but
such information will not always be available. Prior information on the types of errors in the data is useful but not necessary
to identify some of the errors made by participating CSs. When prior error information is known, the model reliably infers
the same five errors, even when the uncertainty of this information is high (i.e. high variance assigned to the Gaussian prior
distributions). When no prior information is known about the potential types of errors present in the data (i.e. $\alpha_\varepsilon \sim \mathcal{N}(1, 100)$,
$\beta_\varepsilon \sim \mathcal{N}(0, 100)$), the model reliably infers the none error type and splits the meniscus error into two error types—a 2-mm
meniscus error and a 3.8-mm meniscus error. The two remaining error types identified are variations on the unknown error
type with relatively low $R^2$ values, 0.79 and 0.09 compared with $R^2$ values of 1.0 for the none and meniscus errors. The model
may fail to identify the unit error type, because it occurs in only 0.7% of submitted observations. Multiple local optima exist for
the error types, and the model may fail to identify all unique errors if no prior information on the errors is known. Regardless
of whether error information is known previously, model evidence indicated that four communities and five error types best
capture the variance in the data. When the priors are vague, the model may require many more iterations (possibly up to 100)
to converge.

There is also some variation in the inferred posterior distributions that is based on how the algorithm is initialized, but the
variation is not statistically significant (p>0.05, per a two-tailed student's T-test). Changing the algorithm initialization during
inference minimally affects the posterior distributions of the error types. For example, with a different initialization, the $\alpha$, $\beta$,
and $\tau$ of the slope outlier change from (10.31, -0.69, 1.5) to (10.31, -0.24, 1.5). The $\alpha$, $\beta$, and $\tau$ values of the remaining error
types are more consistent than the slope outlier type, regardless of how the algorithm is initialized.

### 4.2   Number of communities and error types

Model evidence indicated that there are four communities and five error types present in the data, given the model structure
(see Fig. 3). In comparison, S4W-Nepal identified four error types in the data based on visual inspection of the submitted
observations. The inference model, however, is a much more powerful tool for uncovering nuances in the data than graphi-
cal techniques. Therefore, the number of communities and error types inferred from the model were used for the remaining
analysis.





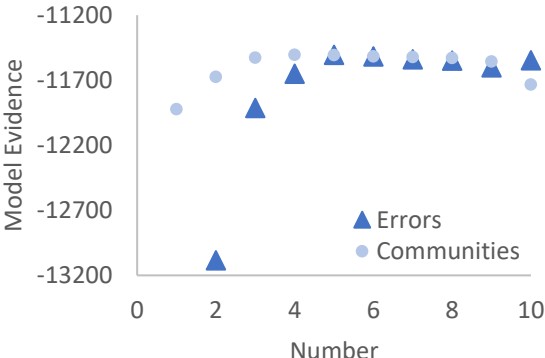

**Figure 3.** Model Evidence for selecting the number of communities and number of error types present in the data given the model structure. Note that model evidence for communities was calculated using five errors. Model evidence for error types was calculated using four communities.

## 4.3 Error analysis

Parameters for the error-specific linear regressions were inferred for the five error types in the submitted rainfall observations (see Table 1 and Figure 4). The inferred parameters included the mean and precision, $\tau$, of the Gaussian distribution, where
the mean is based on a linear regression of $\alpha$, $\beta$, and $\vartheta$ as shown in Equation 1. Four of the five error types align well with the error types identified by Davids et al. (2019): none, unit, meniscus, and unknown. The none error type occurs when the submitted observation matches the true value of the rainfall, as determined from the corresponding submitted photo. The posterior distribution of submitted observations inferred to have a none error type has a high precision (55750), because there is no deviation from the $\alpha$ and $\beta$ values across the submitted observation/true value pairs. Every submitted observation inferred
to have no error exactly matches the corresponding true value. Meniscus errors occur when a CS reports the top of a concave meniscus rather than the bottom of the meniscus. Unit errors indicate instances where a CS submitted an observation in units of centimeters rather than millimeters, resulting in a unit error slope, $\alpha$, of 0.10. Unknown errors do not present a discernible pattern that would explain their origin, as indicated by the low inferred precision (0.01) for this error type. Figure 4 shows that the model-inferred error types are accurate, with only the unknown error type encompassing highly variable submitted
observation/true value pairs.

The inference model identified one error type that was overlooked during the Davids et al. (2019) analysis of errors in the Nepal citizen science data: slope outliers. Slope outliers signify a case where the CS's reported observation was approximately ten times greater than the true value evident in the accompanying photograph of the rainfall gauge. The underlying cause of outlier errors is unclear, but these outliers can likely be attributed to typos (e.g. adding an additional zero) or a mistake made
by reading the gauge from the wrong direction (e.g. top down). Of the 6,091 observations included in the training data, only two were labelled as slope outliers.





**Table 1.** Inferred regression parameters for the different error types

| Error type | Slope, $\alpha$ | Intercept, $\beta$ | Precision, $\tau$ |
|---|---|---|---|
| None | 1.00 | 0.00 | 55750.04 |
| Unit | 0.10 | 0.07 | 36.89 |
| Meniscus | 1.00 | 2.54 | 1.74 |
| Unknown | 0.97 | 2.37 | 0.01 |
| Slope Outlier | 10.31 | -0.69 | 1.50 |

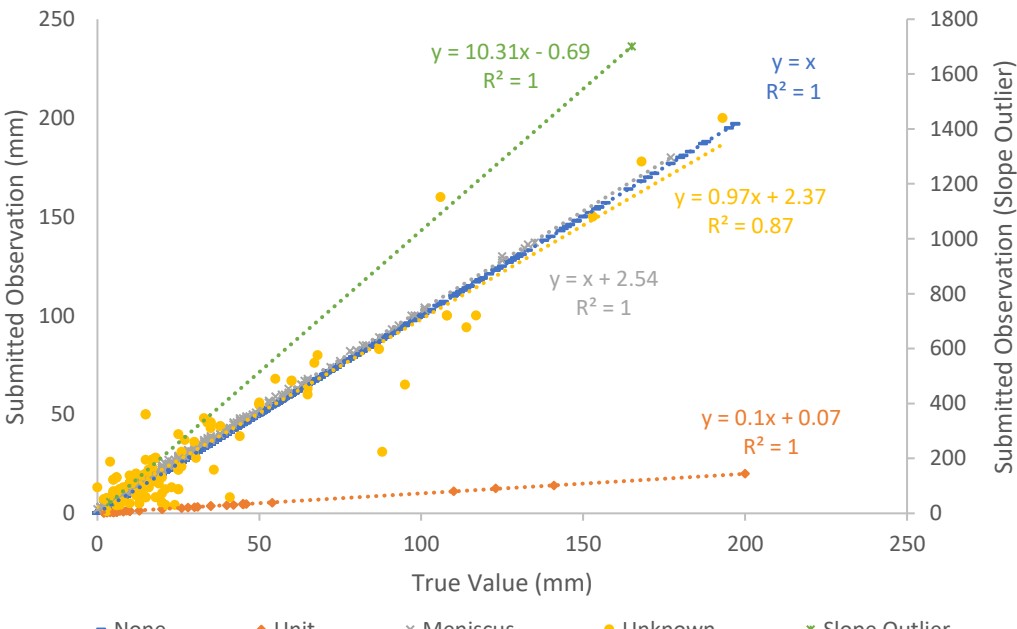

**Figure 4.** Inferred error types for each pair of submitted observation and true value of rainfall in the training dataset. Note that the above plot shows all 6,091 observations used in the training dataset. Few points can be clearly discerned, however, because most (91%) fall along the none error type line and are overlapping elsewhere.

### 4.3.1 Error distribution within communities

The distribution of errors committed by CSs varied depending on the assigned community, as seen in Table 2. Each community was named based on its respective error distribution: Few, Few-MUn, Meniscus, and Unknown Error. The Few community makes very few errors—only 2% of submitted observations are erroneous. Of the erroneous submissions, members in the Few community are most likely to make meniscus or unknown errors (1% each). The Few-MUn community also makes relatively few mistakes but does so at a rate of 5%. Members of the Few-MUn community are almost equally likely to make meniscus errors (3%) and unknown errors (2%). The two other communities, Meniscus and Unknown Error, are much more likely to





**Table 2.** Distribution of errors made by CSs in each community

| Community | None | Unit | Meniscus | Unknown | Slope outlier |
|---|---|---|---|---|---|
| Few (0.47) | 0.98 | 0.00 | 0.01 | 0.01 | 0.00 |
| Few-MUn (0.26) | 0.95 | 0.00 | **0.03** | **0.02** | 0.00 |
| Meniscus (0.20) | 0.80 | 0.01 | **0.17** | 0.02 | 0.00 |
| Unknown Error (0.07) | 0.78 | 0.06 | 0.06 | **0.11** | 0.00 |

*Note*: The probability of each community is shown in parentheses after the community name.
Bold values indicate the most common error type(s) for each community. The probabilities may
not add to 1.0 due to rounding.

submit erroneous rainfall observations. The Meniscus community submits erroneous observations at a rate of 20%. These

observations are largely erroneous due to CSs reading the meniscus of the water incorrectly (17%). Lastly, the Unknown Error community makes the most errors, with 22% of its observations requiring correction. While the Unknown Error community makes primarily unknown errors (11%), meniscus (6%) and unit (6%) errors still represent a large portion of the erroneous submissions. Members of the Unknown Error community are prone to making a wide variety of errors.

The Few community members may have a high degree of scientific literacy; more than 97% of Few community members

have at least a Bachelor's degree. The Few-MUn community members may also have high scientific literacy but occasionally make mistakes. CSs that were initially error prone but were able to correct their misunderstandings based on the feedback provided by S4W-Nepal may also be assigned to the Few-MUn community. For example, one CS in the Few-MUn community made 3 mistakes in the first 16 submissions, but then submitted 44 observations over the next 1.5 years without making a mistake. The Meniscus community largely misunderstands how to correctly read the depth of water in the rain gauge. The

Unknown Error community has several misunderstandings that cross multiple error types, therefore CSs in this community make a mix of errors.

The distribution of errors within each community is a useful tool not only for selecting which submitted observations might require verification, but also for identifying opportunities to improve or maintain the overall accuracy of submitted observations. Citizen science project organizers can use targeted training to help specific communities improve their performance

(Budde et al., 2017; Sheppard and Terveen, 2011). For example, S4W-Nepal could occasionally send feedback messages to the meniscus community members reminding them to read the rainfall depth from the bottom of the meniscus. As another example, members in the Few community might positively respond to general feedback messages acknowledging their strong record of accurate observations and choose to remain engaged with the program. Knowing the error structure of observations submitted by different communities may help improve the overall effectiveness of citizen science programs.

### 4.4 Community composition

The model grouped CSs into four distinct communities with a unique combination of characteristics and probability of making errors. The Few community is the largest with 47% of CSs in the training group assigned to this community (see Table 2). The Unknown community is the smallest with only 7% of CSs classified into this group. The remaining CSs are grouped into the



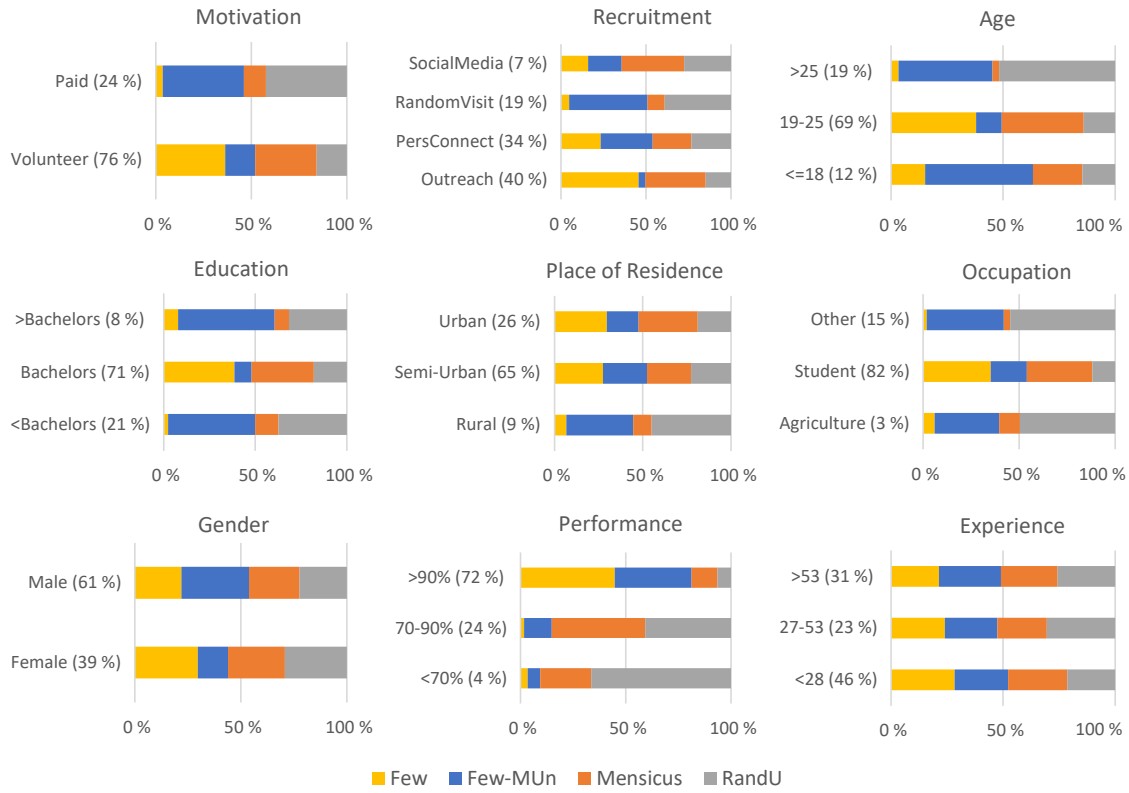

**Figure 5.** Community composition for each characteristic. The percentage of participating CSs with the associated characteristic is shown in parentheses.

Few-Un (19%) and Meniscus (16%) communities. Overall, only 24% of participating CSs are likely to make errors in more than 8.3% of their submitted observations.

The probability that a CS will belong to a specific community depends, in part, on the unique characteristics of that CS. Figure 5 provides the posterior probability that a CS with a particular characteristic would belong to each community, offering insight into the characteristic composition of each community. Singular characteristics may have a large impact on the tendency of a CS to make errors, and therefore to be assigned to a specific community. However, it is also true that any combination of characteristics could contribute to the probability of a CS being assigned to a community. In some cases, CSs are likely to possess a similar combination of characteristics, which surfaces in the community distributions. For example, Figure 5 indicates that CSs recruited during a random visit, older than 25 years of age, holding less than a bachelor's degree, and with an "other" occupation make up 20% of all CSs in the project and have a similar community distribution. While community assignment trends for singular characteristics can be enlightening, the impact of multiple CSs with a similar combination of characteristics must be acknowledged.





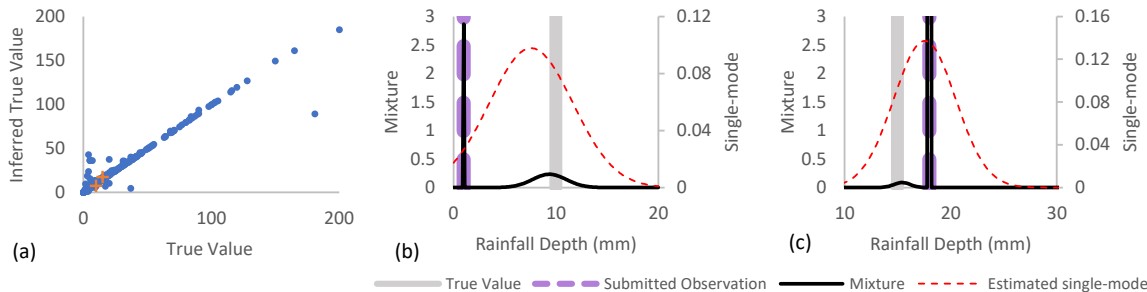

**Figure 6.** (a) The inferred true value is usually a good estimate of the true value of the submitted observation. In some erroneous submissions, the mode of the estimated single-mode posterior is not equal to the true value, however an exact Gaussian mixture of the true value posterior distributions has a local peak at the true value of an observation submitted with a (b) unit error and a (c) meniscus error. The points shown in (b) and (c) are indicated by a plus (+) in (a).

### 4.5 Inferring the true value of a submitted observation

In addition to providing insight into the error structure of the submitted observations and the relationship between CS characteristics and error tendencies, the model provides information about the true value of submitted observations. Testing the model reveals that the model can infer a previously unknown true value based on the value of the submitted observation and

the characteristics of the CS. The inferred true value differs from the actual true value by a median percent error of 0.9%. The standard deviation of percent error is, however, 98.8%. With a wide true value prior distribution (here, 24,000; see Eq. A1), the model has a tendency to over-predict unit errors for a small number of observations submitted with a value of 6 mm or lower which causes the large standard deviation (see Figure 6a). In most cases, the actual true value of the submitted observation falls within the range of the posterior distribution inferred for the true value variable as seen in Figures 6b,c. However, as

Figures 6b,c show, the mode of the posterior distribution is not always a good estimate of the actual true value.

To increase the computational efficiency of an inference algorithm that sometimes needs to consider thousands of variables, expectation propagation approximates a multi-mode posterior distribution with a single-mode distribution (Minka et al., 2018) by minimizing the Kullback-Leibler divergence between the two distributions (Minka, 2005). In many applications, this method works very well. However, here, the mixture distribution covers values ranging from 10% (unit error) of the true value up

through 1,000% (slope outlier error) of the true value. Such a wide range of possible true values results in a predicted true value posterior with high variance and a mode that is occasionally shifted left or right of the true value (see Figures 6b,c).

While the predicted single-mode true value posterior distribution does not always estimate the actual true value of an erroneous submission well, the exact Gaussian mixture posterior often exhibits a local peak at the actual true value (see Figures 6b,c). The mode of the Gaussian mixture posterior usually presents at the value of the submitted observation because of

the high precision associated with the none error type (see Table 1). Only 8.7% of submitted observations have greater than a 20% probability of being erroneous in this example application. Therefore, the inferred error type posterior distribution may be examined in conjunction with the Gaussian mixture posterior to provide additional information on the probability of each error



**Table 3.** Synthetic tests inferring true value from multiple observations submitted for a single event with a true value of 15 mm

| No. obs. | Error types | Inferred Value (mm) | Variance (mm$^2$) | No. obs. | Error types | Inferred Value (mm) | Variance (mm$^2$) |
|---|---|---|---|---|---|---|---|
| 2 | 0, 1 | 14.98 | 6.26E-2 | 2 | 2, 4 | 168.10 | 2.89 |
| 2 | 0, 2 | 15.00 | 1.39E-3 | 3 | 0, 2, 4 | 15.00 | 6.54E-5 |
| 2 | 0, 3 | 14.99 | 1.39E-2 | 3 | 0, 3, 4 | 15.00 | 5.43E-5 |
| 2 | 0, 4 | 153.85 | 3.45 | 2 | 1, 3 | 16.69 | 4.23E-1 |
|  | Different | 15.00 | 9.04E-5 | 3 | 0, 1, 3 | 14.99 | 1.66E-2 |
|  | CS community | 15.00 | 5.94E-2 | 2 | 1, 2 | 17.35 | 5.96E-1 |
|  | combinations | 15.00 | 5.94E-2 | 3 | 0, 1, 2 | 15.00 | 3.27E-3 |
|  | ... | 15.00 | 5.94E-2 | 2 | 2, 3 | 17.59 | 1.91E-1 |
| 3 | 0, 0, 4 | 150.70 | 1.22 | 3 | 0, 2, 3 | 15.00 | 3.29E-3 |
| 4 | 0, 0, 0, 4 | 150.50 | 0.64 |  |  |  |  |

*Note* : Error Types: 0=None, 1=Unit, 2=Meniscus, 3=Unknown, 4=Slope Outlier

type. For example, despite the mode of the Gaussian mixture posterior being located at the value of the submitted observation in Figure 6b, the probability of a none error type is only 0.23, and the unit error probability is 0.73. The Gaussian mixture

posterior and the error type posterior distributions may provide a more accurate representation of the true value of a submitted observation than the approximated single-mode Gaussian posterior distribution.

### 4.5.1 Multiple observations of a single event

If only a single observation of a rainfall event is available, the predicted error type is based on the error types observed during model training. However, analyzing multiple observations of a single rainfall event should improve the accuracy of the inferred

error type and true value of rainfall.

For each of the simulations described below, the model was not given any information about the error types associated with the submitted observations. The model inferred the true value solely based on what it learned during model training. When only one error was made out of two observations submitted, the model predicted the true value every time except for instances of a slope outlier error (see Table 3 column 1). In such cases, the ability of the model to correctly infer the event true value

was related to the error communities of the CSs. Through 12 trials (not shown) with different algorithm initialization and combinations of CSs from the Few-MUn and Meniscus communities, the model correctly inferred the true value only twice. However, the model was able to infer the true value if one submitted observation had a slope outlier for other combinations of CS communities (see Table 3 column 1). If one slope outlier observation was paired with two or more correct observations, the model consistently failed to infer the correct true value. The low probability of a slope outlier combined with the relatively high

probability of unit and meniscus errors cause the model to infer the slope outlier as a meniscus error and the correct observations





as unit errors. When one slope outlier error was paired with another error, the model required an additional correct observation to accurately predict the true value (see Table 3 column 2). For the best performance, the slope outlier error needs to be paired with at least one other erroneous observation and a correct observation. When two errors were made out of two observations submitted, the model often failed to correctly predict the true value. However, when a third observation without an error was
included, the model predicted the true value every time (see Table 3). Overall, the model inferred the correct error types when the inferred true value was also correct.

For instances when multiple observations of a single event are submitted, at least one error-free observation is likely necessary to ensure that the model predicts the true value with minimal uncertainty. When multiple erroneous observations are submitted, the model performs best when at least one correct observation is submitted of that same event. Given that over
90% of submitted observations do not have an error, it is unlikely that an erroneous observation would be submitted without a complementary error-free observation, assuming that additional CSs are active.

### 4.5.2    CSs with unknown characteristics

As CS programs expand, recording complete characteristics data for each participating CS may become challenging. The model's ability to infer the correct community for CSs with unknown characteristics and the correct true value for the ob-
servations they submit was investigated. The characteristics for each unknown CS were selected from a discrete distribution estimated from the characteristics data of CSs observed during training. The prior distribution of the community, $PCom$, was set to a discrete distribution equal to the overall community posterior distribution of the training set. The community for each CS and the true values of their submitted observations were inferred and compared to the communities and true values inferred when the characteristics were known precisely, but the community was also unknown.

The model performed well while inferring the community of unknown CSs and the true values of observations submitted by unknown CSs. Communities of CSs with known characteristics were correctly predicted 0.9% more than CSs with unknown characteristics. The coefficient of determination between the actual true values and predicted true values was 0.015 higher for known CSs than for unknown CSs. While the predicted true values for known and unknown CSs were similar, the uncertainty of the true values predicted from observations submitted by unknown CSs was higher. The average variance of the inferred true
value posteriors was 140.2 mm$^2$ for unknown CSs and 125.6 mm$^2$ for known CSs. Overall, the value of submitted observations has greater influence on the inferred true values of rainfall than the characteristics of the associated CS. While knowing the characteristics of all CSs increases the accuracy of predicting the true value of submitted observations, it is not essential.

### 4.6    Limitations in application

While the model has potential for adaptation to a wide variety of citizen science programs, it has limitations. For example,
the model is data intensive, because a large dataset is required for training and testing the model. This limits its utility for small-scale or newly developed citizen science programs. In addition, a record of erroneous data is required for training the model, which must be identified and corrected by the citizen science program. This may require a large effort and, depending on the type of data collected, may be difficult to achieve. It could be interesting to investigate to what extent the model can be



trained without the availability of error-free ground truth data. For example, Schoups and Nasseri (2021) showed that fusion of multi-source data with unknown noise and bias (in their case, water balance data from remote sensing) is possible in the absence of ground truth data. Lastly, the model design requires that CSs are registered with the program, and that submissions can be linked to registered individuals. This is not the case for all citizen science programs- some do not require registration and some do not track the submission record of their participants. The model can be implemented for quality assessment in many citizen science programs, but the model is not universally useful or without limitations.

## 5 Summary and conclusions

This study developed a probabilistic model to investigate the type and frequency of errors in citizen science data. The model assigns CSs to a community based on the characteristics of the CS and their tendency to submit erroneous observations. This helps to target manual corrections of CS data. The model then infers a posterior distribution of the true value of a submitted observation from the value of the observation and the community of the participating CS. Designed thus, the model can be adapted to a wide array of citizen science datasets.

Analysis of the error structure in CS rainfall observations revealed that individuals can be characterized by one of four error patterns: not error prone, mostly not error prone, meniscus error prone, and random or various error prone. While the Bayesian inference model developed here used communities to relate CS characteristics to error tendencies, the magnitude and type of errors committed is the crux of every community assignment. The distribution of characteristics within each community is useful for investigating potential reasons for making errors rather than for identifying individuals who might be particularly error prone.

The Bayesian inference model developed using Infer.NET's software framework uncovered five error types and their probability distribution within each of the four error-based communities. The community assignments are a useful tool for discerning which CSs are more likely to submit erroneous observations that require further review. In addition, community-specific training and feedback messages may be a powerful tool for increasing the quality and frequency of submissions.

The Bayesian probabilistic model was often able to predict the true value of a submitted observation, and the model extrapolated useful error probabilities for each observation. These error probabilities, in conjunction with the model's inferred error-specific regression and precision parameters, can be used to calculate a Gaussian mixture distribution that provides more information about the probable true value of submitted observations than Infer.NET's single-mode true value prediction. As citizen science programs expand to include multiple participants submitting observations of a single event, the model's ability to predict the true value for that event will likely increase.

As a graphical, assumption-based Bayesian inference model, the citizen science error model presented here has potential for adaptation to other citizen science programs with diverse data types. The implementation of error-based communities provides a simple, yet effective method for tracking changes in the types and frequency of errors committed by CSs. The communities also provide opportunities for targeted re-training and feedback to improve citizen science data at the point of collection,





rather than at the point of correction. Improving the quality of citizen science data at every step enables increasingly more CS-supported decision-making and scientific discoveries.

## 6 Future work

Testing and refinement of the model will continue as new citizen science datasets that meet the minimum requirement of having
a CS observation coupled with a known true value are discovered. Some citizen science datasets can be analyzed directly with the model formulation presented here, like Paul et al. (2020) who compare rainfall observations collected by secondary students with co-located automatic rain gauges. Others may require some adjustments to the model due to special features of the data, like the censored stream stage data collected by CrowdWater. The CrowdWater Application collects CS observations of stream stage and the CrowdWater Game crowdsources the true value of the submitted stage observations (Seibert et al., 2019; Strobl
et al., 2019). In addition to testing the model further with new datasets, the model may be improved, for example, by deriving prior distributions from remotely sensed observations.

*Code and data availability.* The dataset analyzed for this study can be accessed in the Supplementary Material published by Davids et al. (2019). The source code developed for this research will be made available via GitHub before this manuscript is published.

## Appendix A: Prior and Posterior Distributions

The prior distribution for the true value of each event ($\vartheta_e$) was a Gaussian distribution with a mean equal to the mean of the entire true value dataset ($\mu_\vartheta$), and a variance equal to four times the variance of the entire true value dataset ($4\sigma_\vartheta{}^2$; i.e., twice the standard deviation). Here, the true value prior was set to a Gaussian distribution with a mean of 15 and a variance of 2400.

$$\vartheta_e \sim \mathcal{N}(\vartheta_e|\mu_\vartheta, 4\sigma_\vartheta{}^2), \tag{A1}$$

The prior distributions for the $\alpha$ and $\beta$ parameters in Eq. 1 were set to a Gaussian distribution parameterized by mean and
variance. In the S4W-Nepal case study, the $\alpha$ and $\beta$ mean and variance were informed by the mean and variance of a series of slopes and intercepts from linear regressions fit to subsets of ($\vartheta, O$) pairs corresponding to error types identified by Davids et al. (2019). Davids et al. (2019) only identified four error types, whereas the model evidence indicated 5 error types were present in the S4W-Nepal dataset. Therefore, in the case study presented, the priors for the first 4 error types are informative and the prior for the last error type is noninformative.

$$\alpha_n \sim \mathcal{N}(\alpha_n|\mu_\alpha, \sigma_\alpha^2) \tag{A2}$$



where $\mu_\alpha$ = (1, 0.1, 1.002, 0.9, 7), and $\sigma_\alpha^2$ = (0.5, 0.5, 2, 50, 70) for the S4W-Nepal case study. Note that the $\sigma^2$ values used are larger than calculated to provide a wider prior distribution. And,

$$\beta_n \sim \mathcal{N}(\beta_n|\mu_\beta, \sigma_\beta^2) \tag{A3}$$

where $\mu_\beta$ = (0, 0.02, 2.3, 4.2, 3), and $\sigma_\beta^2$ = (0.5, 0.5, 0.2, 50, 30) for the S4W-Nepal case study. Similarly, the $\sigma^2$ values used
are larger than calculated to provide a wider prior distribution.

The prior distributions for the $\tau$ parameter in Eq. 1 were set to a Gamma distribution parameterized by shape ($A$) and rate ($B$). In the S4W-Nepal case study, The $\tau$ shape and rate for the first four $\varepsilon$ error types were informed by a Gamma distribution fit to observations that corresponded to the four error types identified by Davids et al. (2019). The shape and rate for the remaining error type was selected randomly, since there was no information available regarding this error prior to training the
model.

$$\tau_n \sim \mathcal{G}(\tau_n|A, B) \tag{A4}$$

where $A$ = (0.25, 0.75, 1.5, 0.5, 15), and $B$ = (0.05, 0.25, 0.05, 0.01, 10) for the S4W-Nepal case study.

Finally, prior distributions for the various probability vectors in the model, i.e. $\mathbf{PChar}_c$, $\mathbf{PCom}_s$, and $\mathbf{PErr}$ were all set to uniform Dirichlet distributions, reflecting a lack of knowledge on these variables.

Putting everything together, Equation A5 gives the posterior distribution for the model. The posterior is obtained by writing the joint distribution over latent variables $\mathbf{X} = (\mathbf{PChar}_c, \mathbf{PCom}_s, \mathbf{PErr}, \vartheta, \varepsilon, \gamma, \alpha_n, \beta_n, \tau_n)$ and observed variables $\mathbf{D} = (\mathbf{Z}, \mathbf{O})$, followed by conditioning on the observations. Here, $N$ is the number of error types present in the CS data.

$$
\begin{aligned}
p(\mathbf{X}|\mathbf{D}) \propto &\prod_{s=1}^{S}\prod_{c=1}^{C} Dis(Z_{s,c}|PChar_c[\gamma_s]) \\
&\prod_{s=1}^{S} Dis(\gamma_s|PCom_s) \\
&\prod_{s=1}^{S}\prod_{e=1}^{E} Dis(\varepsilon_{s,e}|PErr[\gamma_s]) \\
&\prod_{s=1}^{S}\prod_{e=1}^{E}\prod_{n=1}^{N} \mathcal{N}(O_{s,e}|\alpha_n\vartheta_e + \beta_n, \tau_n)^{\delta(\varepsilon_{s,e}-n)} \\
&\prod_{n=1}^{N} \mathcal{N}(\alpha_n|\mu_\alpha, \sigma_\alpha^2)\mathcal{N}(\beta_n|\mu_\beta, \sigma_\beta^2)\mathcal{G}(\tau_n|A, B)) \\
&\prod_{e=1}^{E} \mathcal{N}(\vartheta_e|\mu_\vartheta, 4\sigma_\vartheta^2)
\end{aligned}
\tag{A5}
$$



where the first four lines correspond to eqs. 1-4 in the paper (and to the four factor boxes shown in the factor graph, Figure
1, replicated over the plates that contain them), and the last two lines denote regression parameter and true value priors (not
explicitly shown in the factor graph). For simplicity, the priors for **PChar**$_c$, **PCom**$_s$, and **PErr** are not explicitly shown in the
posterior equation, since they are all uniform Dirichlet distributions and, as such, are absorbed in the proportionality constant.

*Author contributions.* JAE contributed to conceptualization of this study, methodology, software development, and writing of the manuscript.
GS contributed to conceptualization of this study, methodology, software development, and writing of the manuscript. JCD contributed to
collecting and curating the data and editing of the manuscript. NVdG contributed to conceptualization of this study and editing of the
manuscript.

*Competing interests.* The authors declare that they have no competing interests.

*Acknowledgements.* This research has been supported by the National Science Foundation, Division of Graduate Education (grant no. DGE-
1333468) and the Dutch Research Council. Data collection and quality control was supported by the Swedish International Development
Agency (grant no. 2016-05801) and by SmartPhones4Water (S4W). The authors would like to thank S4W's Saujan Maka for instrumental
guidance.



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
