# Peer review of "A Bayesian model for quantifying errors in citizen science data: application to rainfall observations from Nepal"

_EGUsphere, 2023_

## Author Comment (AC1)

**Reviewer 1**

**MAIN COMMENTS**

**- I very much enjoyed reading this paper, which will be (subject to minor revision) a valuable addition to the relatively new field of error bracketing in citizen science datasets. Specifically, Eisma et al. investigate the ability of one machine learning technique to analyse errors in quantitative hydrological data, which goes well beyond more simplistic analyses typical of qualitative data in e.g. biological science. The identification of individuals / communities that are especially "error prone" is especially useful.**

We appreciate your thoughtful review of our manuscript.

**- I felt as though the Introduction could be restructured slightly to focus immediately on citizen science (in hydrology) i.e. "it is popular and increasingly widespread because xxx, but there are many issues that impede its roll-out everywhere including xxx (e.g. lack of trust, incentivisation, lack of continuing engagement, and demonstrated errors / imprecision relative to more traditional monitoring methods". It seemed like a bit of a jump to couch the first sentence in terms of climate change.**

We started the manuscript with a discussion of how citizen science might impact the lives of those who may benefit from the additional data to cast a wider net of interested readers. However, we agree with your point that it is not well connected to the rest of the content. We re-assessed and decided that starting with citizen science more generally is sufficient to engage potentially interested readers. The first paragraph has been removed. The paper now starts with:

*Citizen science programs, organized efforts to collect scientific data in collaboration with members of the public, have become increasingly popular as advances in technology have made the data collection and submission process more accessible (Bonney et al., 2009; Newman et al., 2012). However, some scientists, policymakers, and workers in federal water bureaucracies continue to question the quality of data submitted by members of the public and have yet to accept the legitimacy of scientific discoveries advanced by citizen scientists (CSs) (Hunter et al., 2013; Paul et al., 2018; Riesch and Potter, 2014; Sheppard and Terveen, 2011).*

**- There is another slight jump in the narrative around line 49, where the text moves from citizen science background & contrasting techniques for (qualitative) error removal, to machine learning and GLMs. The paper is largely focused on elaboration of the models, but I suggest linking better the two halves of the Introduction at this point in the text. Perhaps it would be useful to include a few new lines / paragraph on error detection using machine learning in a broad sense (i.e. not restricted to citizen science datasets).**

Thanks for pointing this out. We agree that the transition needs to be more gradual. We added a sentence to help with this to the end of the paragraph describing techniques for error removal (L40):

*Machine learning algorithms have shown promise as a useful tool to increase the utility of citizen science datasets by making it easier for citizen science programs to identify and describe potential errors.*

**- The paragraph after your research questions could probably be excised as it reads like a summary / Conclusions of the study. Alternatively, you could list the basic structure of the article here (e.g. "In Section 2 we describe the design of our probabilistic model …")**

We understand your point. In this section, we were trying to map the research questions to the different parts of the paper to help guide readers. In attempting to do so, we were more expansive than necessary. The paragraph in question has been edited and now reads:

*A probabilistic graphical model was developed to address these questions based on assumptions about the probabilistic relationships between CSs, their characteristics, and types and magnitude of their errors. The model design (research question 1, see Section 2) includes a mixture of linear regressions sub-models relating true and observed values and includes an unknown number of linear regressions. The model also includes a probabilistic sub-model relating CS characteristics to error types. The model was applied to investigate its utility (see Sections 3 and 4), including the capabilities of the model in identifying erroneous observations and predicting the true value of submitted observations and the impact of multiple observations of a single event on model performance (research question 2, see Section 4.5) and model performance when observations are submitted by CSs with unknown characteristics (research question 3, see Section 4.5.2).*

**- You could address (in the Introduction) why rainfall data were chosen for the investigation (i.e. why not streamflow, or soil moisture, or temperature …? And why might the water cycle be a good place for citizen science datasets to be interrogated) – I suspect this has more to do with data availability and access rather than anything more technical (e.g. representative error distributions), but I think it should be addressed. Coming into Section 2.1, rainfall data are mentioned for the first time since line 38, and feels like an afterthought.**

This information has been added to the beginning of section 3, where we describe how the model was applied to a CS dataset. We are choosing to keep the discussion of rainfall data to the model application section to keep the model development independent of the CS dataset. The beginning of section 3 now reads:

*The model developed in the previous section was applied to citizen science rainfall data collected through the S4W-Nepal program. Rainfall data were chosen to test the model, because rainfall is fairly simple to measure and report and is thus the focus of many citizen science programs (Tipaldo and Allamano, 2017). The water cycle offers a great opportunity to interrogate CS datasets, because water is a ubiquitous natural resource that is relatively well-monitored, providing rich datasets against which to compare CS observations.*

**- Related to previous comment: the beginning of Section 3 is focused on data/background and should come earlier, before the model development of Section 2 (and possibly in the Introduction – notably the passages on the study area). Do you need Section 3.2? I suggest simply referring readers to Davids et al. (2019) at the end of Section 3.1.**

We purposely organized the manuscript to focus on the model development independent of the type of citizen science data we tested the model with. We did this to highlight that the model could be applied to multiple types of data and that the concepts underlying the model do not depend on the rainfall data.

We agree that section 3.2 is largely redundant of Davids et al. (2019). We removed section 3.2, as suggested. Section 3.2 is now summarized at the end of Section 3.1 as follows:

*The S4W-Nepal data used in this model application includes CS rainfall data, CS characteristics, and S4W-Nepal-corrected rainfall data. A detailed description of the S4W-Nepal data collection and quality control process can be found in (Davids et al., 2019). The S4W-Nepal program is ongoing and has collected over 24,500 observations from over 265 CSs since 2016. Overall, approximately 9% of submitted rainfall observations are erroneous. Meniscus errors are the most common (58% of errors; records capillary rise), followed by unknown errors (33%), and unit errors (8%; records data in centimeters rather than millimeters) (Davids et al., 2019).*

*CS characteristics from the S4W-Nepal dataset will be used here to relate individual CSs with the likelihood of mistakes in the data they submit. All CS characteristics recorded by S4W-Nepal, regardless of pre-existing evidence that a characteristic is significantly correlated with CS performance, are included in the model. The model will determine the relative importance of each CS characteristic in defining mistake tendencies while inferring the community groups.*

**- The use of "communities" (e.g. line 96) might be slightly confusing to readers more attuned to hearing it in terms of community-led programs. You could insert a caveat / clarification here that "communities" will be used in a statistical sense.**

Thanks for pointing this out. A clarification has been added in line 94:

*The communities identified by the model are purely statistical in nature and do not represent a physical entity or space.*

**- Paragraph starting on Line 110 – the allocation of citizen scientists to a static single community is a huge simplification (necessary for the modelling), but this should probably be spelt out more explicitly beforehand e.g. in the Abstract.**

Thanks for indicating the need for additional clarity. The term "static" has been added to the abstract in two places and the term "single" has been added once to emphasize the fixed nature of the communities:

*The model assumes that: (1) each CS observation is subject to a specific error type, each with its own bias and noise; and (2) an observation's error type depends on the static error community of the CS, which in turn relates to characteristics of the CS submitting the observation. Given a set of CS observations and corresponding ground-truth values, the model can be calibrated for a specific application, yielding (i) number of error types and error communities, (ii) bias and noise for each error type, (iii) error distribution of each error community, and (iv) the single error community to which each CS belongs. The model, applied to Nepal CS rainfall observations, identifies five error types and sorts CSs into four static, model-inferred communities.*

**- Paragraph starting on Line 362 – this is really exciting and perhaps one of the most important outcomes of the research. As such, I think you should place it more in the foreground (perhaps earlier in the Discussion, as well as a sentence in the Abstract / Conclusions?). Tailoring error messages and ways of improving observations to**

**separate communities would be a significant step-change in enhancing the quality of citizen science data, and therefore their uptake.**

While we agree that this model presents an exciting opportunity, we do not think it can be moved earlier in the discussion. The discussion is organized to mimic the flow of running simulations: algorithm initialization questions, inferring number of communities and error types, and then prevalence and pattern of errors, where you find the comment in question. We do agree that it can be emphasized more in the paper and have made some adjustments.

A clause has been added at the end of the abstract: *and provides an opportunity for targeted re-training of CSs based on mistake tendencies.*

A sentence in the conclusions has been edited and now reads: (L468) *In addition, training and feedback messages tailored to a community's error tendencies may be a powerful tool for increasing the quality and frequency of submissions.*

**MINOR COMMENTS**

**- Line 18 - "measures that sometimes save hundreds, if not thousands of lives" - could you add a ref here? Seems a bit vague**

This text has been removed in response to an earlier comment.

**- Line 20 on institutional capacity – this is a good point but could use a citation**

This text has been removed in response to an earlier comment.

**- Could you fix the CS acronym? Defined as CS in the Abstract but CSs in the main text**

We agree that the use of CSs is a bit odd, but we defined citizen scientist (singular) as CS in the abstract, and citizen scientists (plural) as CSs in the main text. The first instance of citizen scientist(s) in the main text is plural, and we chose to define the abbreviation at the first instance. We do not see a way to fix this and still be grammatically correct.

**- Line 40 – on time and effort spent on QC varying widely – repeats the beginning of that paragraph (Line 33)**

The sentence at the end of the paragraph was removed.

**- Line 63 – "Error modelling has only been employed … in a limited manner" – could you include a citation here?**

Two citations were added (Bird et al., 2014; Kosmala et al., 2016)

**- Beginning of Section 3 – to my mind the real value of citizen science rainfall observations is to capture rainfall extremes that are missed by satellite estimates. You could mention that Nepal has a lot of these extremes, as well as dramatic spatial variations, due to the interaction of the Monsoon with topography.**

The beginning of section 3 has been updated and now reads: *Satellite rainfall data is available worldwide but at a resolution too low to capture rainfall extremes, which may be hyperlocal and rapidly evolving (Stampoulis and Agnagnostou, 2012). Rain gauge observations submitted by CSs have immense potential to record these extremes and increase the scientific community's understanding of rainfall.*

**- Line 195: "rain gauges are notoriously inconsistent" – could you elaborate – in what way?**

Two sentences were added to clarify:

*A wide array of naturally-occurring and equipment-based factors contribute to the inconsistency of rain gauges. For example, erroneous rain gauge measurements may arise from gauge height, splash, wind, poor gauge installation (location and technique), and clogging of gauge inlets among others (Davids et al., 2019).*

**- Line 216: "100 mm diameter clear plastic bottle" rather than "clear plastic bottle with a 100 mm diameter"**

This section has been removed entirely in response to a previous comment.

**- Line 233: might be worth explicitly defining "meniscus errors"**

This section has been removed entirely in response to a previous comment. Meniscus errors are described in L209-210: "Meniscus errors are the most common (58% of errors; records capillary rise)..."

**- Line 337: I did not fully understand this part of the Discussion on slope outliers? They seem fairly insignificant in the statistical sense to me?**

You are correct. Slope outliers occur very rarely. We have reduced the discussion on slope outliers and moved it to the previous paragraph where the other error types were discussed. In response to a comment from another reviewer, we have also added a bit of information on slope outliers in section 4.2. The discussion of slope outliers in section 4.3 now reads:

*Slope outliers signify a case where the CS's reported observation was approximately ten times greater than the true value evident in the accompanying photograph of the rainfall gauge. The underlying cause of outlier errors is unclear, but these outliers can likely be attributed to typos (e.g. adding an additional zero) or a mistake made by reading the gauge from the wrong direction (e.g. top down).*

---

## Author Comment (AC2)

**Reviewer 2**

**General Comments**

**The manuscript "A Bayesian model for quantifying errors in citizen science data: application to rainfall observations from Nepal" presented by J. Eisma et al. introduces a graphical Bayesian inference model to (1) analyze and categorize various error types present in citizen science data and (2) classify the citizens into groups (communities) according to the error distribution within each group. By considering specific error types, the model allows a comprehensive understanding of the error structure of crowdsourced data.**

**The model was applied to real crowdsourced rainfall observations collected within the project SmartPhones4Water in Nepal. The model identified five distinct error types and classified the citizens into four inferred communities based on their error patterns. Leveraging this information, the model enables the identification of observations that require further verification, reducing the burden of data validation on human efforts by employing machine-learned algorithms.**

**I enjoyed reading the manuscript and acknowledge the potential of using a Bayesian model as a novel approach to improve the efficiency and accuracy of data validation in citizen science. The findings underscore the importance of well-defined error structures in citizen science data and demonstrate the value of a graphical Bayesian inference models in understanding and harnessing such data effectively, which becomes more and more relevant with the increasing amount of crowdsourced data. Overall, the manuscript is well structured and contributes relevant insights to the emerging field of citizen science data collection and, subsequently, the use of such data for further research. I recommend considering this manuscript for publication in HESS with minor revision.**

We appreciate your thoughtful review of our manuscript.

**Specific Comments**

**L24: Consider removing the word "traditional" in front of scientists. What are "non-traditional" scientists – and – in general, all scientists should be concerned about data quality from whatever source.**

Good point. The word "traditional" has been removed.

**L229: Are the data also checked/calibrated by automatic rain gauges installed according to certain quality assurance standards? If so, this section may need to be briefly expanded to include a comparison of the overall data quality between CS data and automatically collected data. However, as the overall quality of CS data is not the focus of this manuscript, this comparison is not critical.**

We did not do this comparison, but the original data paper (Davids et al., 2019) compared the rain gauges used with a few different standard rain gauges. The line to which you are referring has been removed, and the Davids et al. (2019) paper is referenced for a detailed description of the dataset. See section 3.1.

**L252: As an additional filter for the data, the authors set a maximum limit of 540 mm of rainfall per day. My concern with this limit is that citizens may not be able to record such an event because the rain collector would overflow. In this case, the maximum amount of precipitation that can be measured by a CS station per day/measurement might be a more realistic limit. The authors should also report the number and percentage of data points that exceeded the upper (and lower) limit.**

This is a great point. The S4W-Nepal rain gauges had an upper limit of 200 mm. We changed the value in the model and re-ran the simulations. It did not change the results for this particular dataset. The start of section 3.3 has been edited to read:

*Before training and testing, an additional assumption was incorporated due to the nature of rainfall data: the inferred value of rainfall was assumed to be between 0 and 200 mm. Rainfall events cannot result in negative rainfall, and 200 mm is the maximum rainfall depth that can be recorded in a single measurement using the S4W-Nepal rain gauges per Davids et al., (2019). The CS rainfall observations analyzed here include no observations below 0 mm and two observations (0.03%) greater than 200 mm. One could include overflow as another type of error but given the rare occurrence, that was not included here.*

**L320: Maybe name the error type that was introduced with the model here (slope outliers). It is mentioned in section 4.3, but I was missing this information in this section. It might be also valuable to expand this section slightly to explain why "slope outliers" have been identified as an error type. When looking at the distribution of errors made within the communities (Table 2), slope outliers never occurred. The relevance of this error type remains unclear to me.**

The name and a brief description of the significance of the error type have been added to section 4.2.

*The model inferred one previously unidentified error type in addition to the four error types that were identified by S4W-Nepal's visual inspection of the submitted observations (Davids et al., 2019). The additional, model-inferred error type, named slope outlier, is significantly different from the other identified error types (see Table 2) and only occurs twice in the training and testing data. Each identified error type will be explored more fully in the next section.*

**L344: I would recommend using a different term for the Few-MUn group. The "few group" also makes only Meniscus and Unknown errors – similar to the Few-MUn group. The only difference is the overall amount of errors (2 % vs 5%). Hence, the groups could be named according to the amount of errors (such as p2 and p5 group, or minor and few, etc.). This would also improve the readability of the manuscript.**

The Few-MUn community has been renamed to the Few+ community throughout the manuscript. For example, see Table 2 and Figure 5.

**L457: The authors mention that a set of erroneous data is required to train the model and that these data need to be identified and corrected by the CS program, which can be a significant effort. Other studies have shown that this task could also be done in collaboration with the community (e.g., Strobl et al. https://doi.org/10.1371/journal.pone.022257). It may be of interest to the readers of this**

**study to include some information on this approach here. This is currently listed in Section 6 (Future work) but may fit better within the discussion in Section 4.6.**

Thanks for making this connection for us. CrowdWater has a great solution to the identified limitation. Section 4.6 has been updated and now includes:

*This may require a large effort and may be difficult to achieve, but at least one citizen science program, CrowdWater, has an innovative solution. The CrowdWater Application collects CS observations of stream stage, and the CrowdWater Game crowdsources the true value of the submitted stage observations (Seibert et al., 2019; Strobl et al., 2019).*

**L486: A limitation of this study is that it was only tested with one CS project in one region. The authors should mention this limitation more clearly in the conclusion, as it remains unclear whether the method and model developed will work equally well in different settings.**

L476 has been expanded to read:

*While the results are promising, the model was only tested with one citizen science program deployed in one country. Further testing with datasets from different citizen science programs is required to assess whether the method and model perform equally well. Applying the model to different citizen science datasets may require some of the model assumptions to be tailored to the specific application (e.g., range of acceptable values, censored data, etc.). However, the flexibility of the modelling tool used, Infer.NET, makes it simple to vary the model to suit the specific needs of different CS datasets.*